# Factors influencing the implementation of labour companionship: formative qualitative research in Thailand

Somporn Rungreangkulkij,[1] Ameporn Ratinthorn,[2] Pisake Lumbiganon [iD],[3] Rana Islamiah Zahroh [iD],[4] Claudia Hanson [iD],[5,6] Alexandre Dumont [iD],[7] Myriam de Loenzien [iD],[7] Ana Pilar Betrán [iD],[8] Meghan A. Bohren [iD][4]

For numbered affiliations see end of article.

**Correspondence to**
Dr Meghan A. Bohren;
meghan.bohren@unimelb.edu.au

## ABSTRACT

**Introduction** WHO recommends that all women have the option to have a companion of their choice throughout labour and childbirth. Despite clear benefits of labour companionship, including better birth experiences and reduced caesarean section, labour companionship is not universally implemented. In Thailand, there are no policies for public hospitals to support companionship. This study aims to understand factors affecting implementation of labour companionship in Thailand.

**Methods** This is formative qualitative research to inform the 'Appropriate use of caesarean section through QUALIty DECision-making by women and providers' (QUALI-DEC) study, to design, adapt and implement a strategy to optimise use of caesarean section. We use in-depth interviews and readiness assessments to explore perceptions of healthcare providers, women and potential companions about labour companionship in eight Thai public hospitals. Qualitative data were analysed using thematic analysis, and narrative summaries of the readiness assessment were generated. Factors potentially affecting implementation were mapped to the Capability, Opportunity, and Motivation behaviour change model (COM-B).

**Results** 127 qualitative interviews and eight readiness assessments are included in this analysis. The qualitative findings were grouped in four themes: benefits of labour companions, roles of labour companions, training for labour companions and factors affecting implementation. The findings showed that healthcare providers, women and their relatives all had positive attitudes towards having labour companions. The readiness assessment highlighted implementation challenges related to training the companion, physical space constraints, overcrowding and facility policies, reiterated by the qualitative reports.

**Discussion** If labour companions are well-trained on how to best support women, help them to manage pain and engage with healthcare teams, it may be a feasible intervention to implement in Thailand. However, key barriers to introducing labour companionship must be addressed to maximise the likelihood of success mainly related to training and space. These findings will be integrated into the QUALI-DEC implementation strategies.

## STRENGTHS AND LIMITATIONS OF THIS STUDY

⇒ Labour companionship has important benefits for the woman and baby and is recommended by WHO. This is the first study to understand needs and preferences related to labour companionship and map factors that might affect implementation of labour companionship in Thailand.

⇒ We found that implementation of labour companionship is feasible if labour companions and health workers are well trained on how to best support women and engage with one another. Addressing key barriers to introducing labour companionship can include changes to the physical environment, implementing facility-level policies on labour companionship and context-specific solutions to minimise fears on lawsuits and infection.

⇒ A key strength of our study is the triangulation of qualitative research and facility readiness assessments and mapping of key factors affecting implementation of labour companionship to the Capability, Opportunity, and Motivation (COM-B) model of behaviour change.

⇒ Using the COM-B model to guide analysis, we show how to use our formative research findings to guide intervention design and support a systematic, targeted and theory-based development of implementation strategies for labour companionship.

⇒ While our research was conducted in eight public hospitals across different regions of Thailand, the findings may not be transferrable to all settings in Thailand, as most study hospitals were in urban settings with high caesarean section rates.

## INTRODUCTION

Efforts to improve maternal health globally have shifted in recent years to improving quality of care. A critical component of quality of care is the person's 'experience of care', which the WHO has defined as ensuring that all pregnant people are treated with respect and dignity, have effective communication with health workers, and access to emotional support that meets their needs.[1] Within labour and childbirth

care, supporting women to have a labour companion of their choice present is an effective way to improve women's experiences by providing respectful care and emotional support.[2 3] Labour companionship refers to a person of the woman's choice, who accompanies the woman continuously throughout labour and childbirth; typically, this is the woman's partner or husband, friend or family member.[4] Labour companionship empowers women in several key ways: improving communication between women and health workers, helping women with non-pharmacological pain relief, acting as advocates to help voice the woman's preferences, providing practical support such as massage and hand-holding and providing emotional support as a continuous presence.[4]

Labour companionship has important benefits for both the woman and baby. A Cochrane intervention review analysed the impact of continuous support for women during labour and childbirth from 26 studies conducted with over 15 000 women in 17 countries and found that women with continuous support were more likely to have a spontaneous vaginal birth and less likely to report negative ratings of or feelings about their childbirth experience or to have a caesarean birth.[5] Women with labour companionship also have a shorter duration of labour and better 5 min Apgar scores for their babies. Based on this evidence, WHO recommends that all women have the opportunity to have a labour companion of their choice with them throughout labour and childbirth.[3]

Despite clear evidence of benefit, implementation of labour companionship in health facilities across the world remains suboptimal. A Cochrane qualitative evidence synthesis highlighted several factors affecting implementation, including women and health workers not recognising the benefits of labour companionship, labour companionship viewed as a 'nice to have' but not essential service, physical space constraints on labour wards and thus difficulties to maintain privacy and integrating labour companions into part of the care team.[4]

### Context of labour companionship in Thailand
In Thailand, labour companions are not typically allowed in most public and some private hospitals. Most public hospitals have a policy allowing women's relative to wait outside the labour room, with certain hours allocated to allow relatives or friends to visit the women in the labour room, typically during lunch or dinner time. Anecdotally, some reasons for not allowing labour companionship were the concern about infection risks (even prior to COVID-19) and maintaining the privacy of women, who normally share rooms, especially from other male companions. With increasing access to mobile phones, there are also emerging concerns about pictures and audio video recordings, which may be used in potential litigation cases against medical teams. Similar to the results of the Cochrane review, a quasiexperimental study in eastern Thailand compared the effect of companionship on primiparous women's experiences and found that women with companionship were more satisfied with

their childbirth experiences, but no significant differences in self-reported suffering or ability to cope with labour pain.[6]

### The QUALI-DEC project
In the context of sustained growing caesarean section rates in Thailand, the Ministry of Health and other stakeholders are examining factors underlying the increase and interventions to optimise its use. The QUALI-DEC study: 'Appropriate use of caesarean section through **QUALi**ty **DEC**ision-making by women and providers'[7] aims to design, adapt and evaluate a multifaceted strategy, for the appropriate use of caesarean section in Argentina, Burkina Faso, Thailand and Viet Nam. The QUALI-DEC strategy is designed to combine four key components: (1) opinion leaders to implement evidence-based clinical guidelines; (2) caesarean audits and feedback to help providers identify potentially avoidable caesarean sections; (3) a Decision Analysis Tool to help women make an informed decision on mode of birth and (4) implementation of WHO recommendations on companionship during labour and childbirth.[7] Labour companionship is included as a QUALI-DEC intervention component given the association between continuous support and increased chance of vaginal birth[5] as well as due to emerging evidence that companionship may improve women's experience of care and reduce mistreatment during childbirth.[8 9]

The QUALI-DEC strategy supports the woman to choose any person to act as her labour companion. The QUALI-DEC research team and implementation partners will codevelop and tailor a model for labour companionship in each hospital that includes information on (1) changing hospital policy to allow for labour companionship, (2) establishing eligibility criteria for women and companions, (3) identifying how health workers can help women to choose and train the labour companion, (4) defining how health workers engage with women and companions, how many companions are allowed and when they are present, (5) designing modifications for the physical space to accommodate companions and (6) developing educational tools for companions on how to support women. Based on the formative research conducted among the local stakeholders in Thailand, the aim of this paper is to describe the needs and preferences of women, potential companions and healthcare providers related to labour companionship and to map factors that might affect implementation of labour companionship in Thailand, using a behaviour change model.

### METHODS
This is a formative qualitative study using a health facility readiness assessment and in-depth interviews (IDIs) with women, potential companions and healthcare providers, described in detail in the study protocol[10] and below. In short, the readiness assessment and IDIs explored

**Table 1** Study sites in Thailand

| Hospital # | Region | Type of hospital | # of births per year (2020) | Caesarean section rate (2020) |
|---|---|---|---|---|
| Hospital 1 | Central Thailand | Public hospital | 4431 | 43.6% |
| Hospital 2 | Central Thailand | Public hospital | 4605 | 34.3% |
| Hospital 3 | Central Thailand | Public teaching hospital | 5203 | 48.5% |
| Hospital 4 | Northeast Thailand | Public teaching hospital | 1727 | 42.5% |
| Hospital 5 | Northeast Thailand | Public hospital | 4756 | 43.6% |
| Hospital 6 | Northeast Thailand | Public hospital | 3361 | 49.2% |
| Hospital 7 | Northern Thailand | Public hospital | 5025 | 50.1% |
| Hospital 8 | Eastern Thailand | Public hospital | 3268 | 56.9% |

the needs and preferences of these key stakeholders to introduce labour companionship in each setting. During the analysis, we conceptualised findings from the readiness assessment and IDIs as 'factors potentially affecting implementation of labour companionship', and used behaviour change frameworks to map the findings in order to better understand what is needed to develop effective intervention implementation strategies. This paper is reported according to the consolidated criteria for reporting qualitative research (COREQ) guidance.[11]

Eight hospitals in Thailand were purposively selected for the QUALI-DEC project according to the willingness to participate, programmatic activities, country priorities and geographical representation (table 1). The formative research was conducted in these eight hospitals, where caesarean section rate ranged from 34.3%–56.9%.

### Participants and recruitment

Five groups of participants were identified for this study: (1) pregnant women, (2) postpartum women, (3) a person identified by the woman as someone she would have liked as a companion (potential companions; before birth), 4) potential companions (after birth) and (5) healthcare providers (doctors, nurse-midwives) and administrators or managers. Pregnant women and postpartum women aged 18–49 years who attended antenatal and/or postnatal care at the study hospitals were invited to participate in IDIs, aiming for diversity (mix of urban or rural residence, parity, age and ethnicity—target per facility: 2–3 pregnant and 2–3 postpartum women). Initially, nurse-midwives explored the interest of women during antenatal care or postnatal care visits, and if they were potentially interested in participating, then the research team approached women face-to-face. The pregnant and postpartum women who participated in the study identified a person who they would have liked to be their labour companion ('potential companion') and the research team approached the potential companions face-to-face to participate in an IDI (target per facility: 2–3 potential companions before birth and 2–3 after birth). Typically, the potential companion was already on the hospital grounds. Healthcare providers working on the antenatal, delivery and postnatal wards of the study hospitals and healthcare administrators were contacted by the research

team and invited to participate in IDIs, with considerations for a diverse group based on age, gender and years of working experience (target per facility: 2–3 nurse-midwives, 2–3 doctors, 2 administrators). We prespecified the target sample size for each type of participant to account for the variable contexts and patient populations in each facility. No participants approached refused to participate.

### Data collection

After agreeing to participate and completing a consent form, the research team conducted IDIs in Thai at the respective health facility. IDIs lasted 30–90 min, had no other people present, were audio-recorded and participants received 500 Baht (US$16) compensation for their time. General conversation was initiated prior going to main interview questions to build rapport. Data were collected from July to October 2020. All audio recordings were transcribed verbatim in Thai, complemented with field notes. Deidentified transcripts were stored on a password protected computer. There was no further contact with the research participants after the IDI.

The interview guides were developed based on the implementation challenges identified in the Cochrane qualitative review[4] and covered a range of topics including: (1) values and needs around the childbirth period, (2) prenatal education, (3) preferences and decision-making processes regarding mode of birth and (4) labour companionship (online supplemental appendix 1: interview guide). Interview guides were piloted and refined prior to data collection. This analysis focuses on the labour companionship module.

In addition to IDIs, a readiness assessment was conducted to describe and assess the service delivery context ahead of the intervention implementation and was carried out concurrently with the IDIs (online supplemental appendix 2: readiness assessment). The readiness assessment provides a systematic approach to assessing readiness to engage in the implementation, in order to inform and tailor the interventions in a way best suitable to the local context.[10] Readiness assessments were conducted by members of the QUALI-DEC research team who were professors of nursing, but not employed by the study hospitals. During data collection, the researchers

used a semistructured form to observe the service delivery context in each facility setting related to possibility or barriers for companionship implementation such as the sign for visiting information, and the physical environment in the labour room and postpartum room.[10]

## Reflexivity

The QUALI-DEC research team consists of Thai and international social scientists, nurses, doctors and epidemiologists with maternal health expertise. The research team believed that labour companionship is beneficial for women and families and may help reduce caesarean section rates. The research team was aware of their assumptions and mindful through the study process to mitigate any potentially negative biases that could influence participant responses or interpretations of responses. Six members of research team conducted the IDIs, all were female nursing professors with extensive qualitative experience, no prior relationship with any participants and did not work at the study sites. Prior to starting data collection, the research team underwent a 3-day training on caesarean section globally and in Thailand, QUALI-DEC project and data collection and management.

## Data analysis

Thematic analysis was performed by hand according to the following steps: organising the data; generating categories, themes, patterns; testing emergent hypothesis; searching for alternative explanations.[12] Four members of the research team were involved in the data analysis. First, the researchers repeatedly read the interview transcripts to develop initial codes of the data. Second, the researchers conducted a systematic identification of themes from the codes such as support, being a representative and shorten labour. Third, from the themes and codes, researchers identify emerging patterns from the data, such as benefits of having labour companion. Last, the researchers review the coded data extracts for each theme to consider whether they appear to form a coherent pattern. In this stage, the research team considered how the different themes were similar and different across different participant groups (eg, women and healthcare providers) and explored hypothesis for why these similarities and differences may exist. If we found inadequacies in the initial coding and themes, we revisited the themes again and iterated on necessary changes when needed. For trustworthiness, during data analysis, the findings were discussed among the research team and emergent findings were presented to a representative obstetrician (QUALI-DEC opinion leader) from the study settings. Key themes emerging from the IDIs were combined with data from the readiness assessment to identify and prioritise barriers and to develop potential implications for implementation. Data analysis was conducted in Thai in order to retain the original meaning, and excerpts from the interview transcripts in this article were translated by a bilingual Thai-English translator who is a member of the research team.

The research findings were then conceptualised as factors potentially affecting implementation and mapped to the Capability, Opportunity, and Motivation model of behaviour change (COM-B).[13] The COM-B model theorises that for a desired behaviour to occur (eg, labour companionship), individuals must have the capability, opportunity and motivation to enact the behaviour. Capability refers to factors such as attention, decision-making, knowledge and skills.[13] Opportunity refers to how environments influence behaviour and includes both physical (eg, access to supplies and resources, staffing, infrastructure) and social (eg, team-work, support, practice norms, social and professional identities) contexts.[13] Motivation refers to the internal processes that direct and encourage behaviours to occur or not and includes factors such as perceived benefits, risks and consequences, emotions and priorities.[13] The COM-B model has been widely used in implementation research to improve implementation and to explore barriers and facilitators to changing clinical practice. By identifying factors (eg, barriers and facilitators) that may affect implementation, teams can then design implementation strategies to address these factors and, in turn, optimise the likelihood of successful implementation and potential for scale-up.

## Patient and public involvement

Patients and/or the public were not involved in the design, or conduct, or reporting, or dissemination plans of this research.

## RESULTS

From the eight participating hospitals, a total of 127 IDIs are included in this analysis: 27 pregnant women, 25 postpartum women, 16 potential companions, 8 facility administrators, 18 doctors and 33 nurse-midwives working in maternity care. Table 2 presents the sociodemographic characteristics of women and potential companions. Pregnant and postpartum women's ages ranged from 18 to 42 years, almost all were married or cohabitating with a partner and most were employed. Among pregnant women, about half were nulliparous, including two women who had planned for a caesarean birth. Among postpartum women, at their most recent birth, about one-third had a vaginal birth and two-thirds had a caesarean birth. Almost all potential companions identified by the women were their husbands, except one who was the woman's mother. Table 3 presents the sociodemographic characteristics of healthcare providers. There were 12 men (doctors and administrators) and 59 women (14 female doctors/administrators; all nurse-midwives were women).

## Contextual insights from the readiness assessment

Observations of the eight hospitals during the readiness assessment demonstrated space limitations and crowding

**Table 2** Sociodemographic of participants: women and potential companions

|  | Pregnant women | Postpartum women | Potential companions |
| --- | --- | --- | --- |
| Total number of participants | 27 | 25 | 16 |
| Age (years) | | | |
| 18–24 | 8 | 4 | 0 |
| 25–30 | 9 | 9 | 4 |
| 31–42 | 10 | 12 | 10 |
| 43–59 | 0 | 0 | 2 |
| Marital status | | | |
| Single | 0 | 0 | 0 |
| Married/cohabitating | 26 | 25 | 15 |
| Divorced/widowed | 1 | 0 | 1 |
| Occupation | | | |
| Government officer | 3 | 2 | 0 |
| Business owner | 8 | 5 | 5 |
| Employed (other) | 8 | 11 | 10 |
| Unemployed | 8 | 7 | 1 |
| Parity and planned mode of birth | | | |
| Nulliparous (no planned CS) | 10 | – | – |
| Nulliparous (planned CS) | 2 | – | – |
| Multiparous (no planned CS) | 9 | – | – |
| Multiparous (planned CS) | 6 | – | – |
| Mode of birth (most recent birth) | | | |
| Vaginal birth | – | 8 | – |
| CS | – | 17 | – |

CS, caesarean section.

on the labour ward, typically with multiple beds in the same room, close together and only divided by a curtain.

There are differences in current visiting hours in the labour and delivery wards across the hospitals. Two hospitals (hospitals 5 and 6) limit the visiting hours to three times a day, 1–2 hours in the morning, noon and evening. In contrast, the five other hospitals allow visitors from 11:00 to 20:00 hours, but with limits on the number of visitors and duration of visits. Almost all hospitals allow only one visitor to visit for 15–20 min at a time. There is only one hospital (hospital 8) that allows woman who are in labour to visit the relatives at the ward reception area until 20:00 hours.

Two hospitals (hospitals 5 and 6) provide onsite overnight accommodation for relatives. One hospital (hospital 2) provides accommodation to the relatives only if the woman in labour is under 20 years old. In addition, two hospitals (hospitals 1 and 8) have a room for the relatives to be with the woman in labour until after the birth for extra charge.

Discussions between the research team and clinical staff as part of the readiness assessment suggested that a potential solution for seven hospitals would be to implement labour companionship for some, but not all women. For example, if seven women are in labour at the same time, labour companionship could be piloted with approximately two or three women without compromising care for all women. In these hospitals, it may be possible to make more private space for women during labour, for example, by moving a woman who is in active labour to the corner of the ward and using curtains that are already available. Hospital 5 had serious concerns regarding the seriously limited space that might challenge the implementation of labour companionship.

### Qualitative findings related to labour companionship
The findings showed that, in general, healthcare providers, women and potential companions had positive attitudes about labour companionship. The qualitative findings are grouped in four categories in the subsequent sections: (1) benefits of labour companions, (2) the roles of labour companions, (3) training for labour companions and (4) factors affecting implementation.

#### Benefits of labour companions
Women, companions and health workers expressed similar benefits and challenges of having labour companions, including (1) support, warmth and improved marital

**Table 3** Sociodemographic of participants: healthcare providers

|  | Administrators | Doctors | Nurse-midwives |
|---|---|---|---|
| Total number of participants | 8 | 18 | 33 |
| Gender |  |  |  |
| Female | 2 | 12 | 33 |
| Male | 6 | 6 | 0 |
| Years working in total |  |  |  |
| 1–5 | 0 | 7 | 8 |
| 6–10 | 0 | 5 | 2 |
| 11–15 | 0 | 2 | 1 |
| 16–20 | 1 | 2 | 5 |
| 21–25 | 0 | 1 | 5 |
| 26–30 | 4 | 1 | 4 |
| ≥31 | 3 | 0 | 8 |
| Years working at study facility |  |  |  |
| 1–5 | 0 | 11 | 10 |
| 6–10 | 0 | 1 | 5 |
| 11–15 | 0 | 3 | 4 |
| 16–20 | 1 | 2 | 4 |
| 21–25 | 0 | 0 | 4 |
| 26–30 | 4 | 1 | 3 |
| ≥31 | 3 | 0 | 3 |

relationship, (2) having a representative to communicate with the medical team, (3) perception of clinical benefits, (4) labour companions as witnesses, (5) reduce the nursing workload in emotional support and (6) labour companions may not be helpful. They have noticed the benefits of having the companion included shorten labour duration, to reduce caesarean section, to understand the work of medical team, to reduce the nurse-midwife's workload by being the woman's emotional supporter and to provide opportunity for professional development. These benefits are outlined in the following sections. Recognition of the benefits of labour companionship are important facilitators for the *reflective* and *automatic motivation* domains of behaviour change, as they refer to the conscious thought processes (plans, evaluations) and habits or desires that influence motivation.

## Support, warmth and improved marital relationship
Many women expressed that they feel anxious during the labour and birth. They feared the labour and birth process in the unfamiliar hospital environment. They experienced pain from contractions and worried about their safety and the baby's health. These women believed that having a companion might reduce fear and anxiety:

*It is very nice to have some support. Some people need emotional support, wanting to have some familiar faces around. They looked around - they saw only the strangers. If they could see the mom or the husband, they would have felt some support that at least they have a friend. Having companions*

*is very beneficial.* (Labour nurse-midwife 4, 7 years work experience, hospital 4)

*It's good to have a companion…have someone to talk to while waiting…I would have felt relaxed…But if I were to have someone with me, I would have felt less anxious and forgot the pain a little bit.* (Postpartum woman 4, 35 years, hospital 4)

The participants from all groups said that having a labour companion present during the woman's labour and birth could improve the marital relationship if the husband was chosen to be a companion. The husband and wife could go through the experience of the labour pain, emotional journey together.

*Having my husband as a companion was very good. It's a very good bonding experience before the baby arrives. It's better for our family relationship.* (Postpartum woman 10, 23 years, hospital 5)

*One of the good things about having the companion is that we can support and consult with each other. We can go through it and help each other along the way.* (Husband 1, 29 years, hospital 8)

## Representative to the medical team
Participants described how labour pain can affect the woman's decision-making, perceptions and judgement. Therefore, having a companion during labour who was a family member could be useful to act as a representative to communicate with the medical team. This can improve effective communication of the women's needs and preferences.

*Many times, the patients are in so much pain. We couldn't really communicate with them…They couldn't make sound decisions. If they have a relative who can be their representative, it improves the communication and decision making.* (Obstetrician 3, 10 years work experience, hospital 4)

*Having a companion is a good thing. They can be my representative, if something is wrong. They can get a nurse for me.* (Postpartum woman 4, 35 years, hospital 4)

## Perception of clinical benefits
Some healthcare providers believed that if women had good support, they would be able to manage their pain which, in turn, seem to help shorten the labour duration.

*One of the good things about having the companion is the smooth delivery…For example if the mom is with the patient, the mother might be able to support because the mother has experienced labour before. They can help the patient to follow the medical team's teaching like how to push correctly. The partner can help guide the patient to a successful labour.* (Obstetrician-Administrator 2, 30 years work experience, hospital 4)

Moreover, some healthcare providers believed that when the women had good support, they may manage the pain better than if they did not have a companion. This could result in fewer caesarean births, as some women ask

for caesarean because they no longer wish to tolerate the labour pain.

*Having a companion with the woman seems to help with the surgery request [for caesarean section]. When the women are in labour pain, they will have someone with them to distract from the pain…. Many cases they ask for surgery because they are experiencing labour pain and don't want to wait until the vaginal birth.* (Obstetrician 1, 11 years work experience, hospital 7)

### Labour companions at witnesses

When the women's relatives stay with them throughout labour, they can witness the work of healthcare providers directly. Healthcare workers described that when family members are present, they tended to be more careful while working, which may therefore improve service quality.

*It is like the companions are the quality assurance inspectors. They see how our system works. It is like a two-way communication that we can improve the quality of our service.* (Labour nurse-midwife 13, 34 years work experience, hospital 6)

Healthcare workers also felt the presence of companions could reduce some misunderstanding about medical malpractice, as the companion could witness and understand the work of the medical team which may lead to fewer lawsuits.

*It's beneficial to have a labour companion. If there are any complications during the labour and the delivery, they will see that we try our best. When they see that we are trying the best we can, that might reduce the lawsuits. They have witnessed that we do pay attention. They can participate in the care.* (Obstetrician 2, 3 years work experience, hospital 7)

### Reduce the nursing workload in emotional support

One of the nursing roles is providing emotional support to women during labour. The nurse-midwives also monitor frequency of contractions and provide other nursing care. When there are many women in labour, the nurse-midwives might not be able to provide close attention to every woman, and emotional support in particular can be compromised. Having a labour companion who has been trained on how to support women could therefore potentially reduce the nursing workload.

*It helps reducing my workload.….I try to pay close attention to all my patients. I can do that when I have only a few patients. But when the patient has a labour companion, I feel good that my patients do receive intensive care, even though it's from the companion, not me.* (Labour nurse-midwife 14, 10 years work experience, hospital 6)

### Labour companion may not be helpful

Most participants expressed the benefits of having labour companions. However, there were four women who said that they did not need a labour companion, primarily because they believed that during labour, nobody could help alleviate pain. These women believed that during labour, women tended to have limited attention and negative moods.

*Either way is fine with me, having a companion or not. I am in labour. I will feel pain, no matter I have someone with me or not. Having a companion isn't helping with my pain.* (Pregnant woman 13, 31 years, hospital 5)

Moreover, one husband also said it was not helpful for him to be there. He said it is better for the woman to be with the medical team, and feared to see her suffer.

*I think I will not be a labour companion. I will wait outside the room. I don't want to be in the way of the medical team. I am worried but I don't want to see her crying and suffering.* (Husband 3, 42 years, hospital 3)

### The roles of labour companion

Most healthcare workers said that the women should be the one who select their labour companion. Most women preferred their husbands to be their labour companions, as they think that it will enhance the family relationship, and a few women preferred their mothers as they viewed their mothers' own labour experiences to be beneficial in supporting them. The participants from all groups expressed the roles of the labour companion very similarly, to provide emotional support, massage and support coping with pain, assisting with daily activities, and communicating with the medical team.

*I would like someone who can be around and help out. Someone who holds me when I am in pain. Someone who can help getting things for me when I can't really help myself. It is better than being alone.* (Pregnant woman 10, 38 years, hospital 2)

The health workers also perceived that labour companions could play key roles in supporting them to better care for the women in labour.

*The first thing is to be my support. Other duties can be understanding the labour and delivery process. So that person isn't in panic. If they notice any unusual symptoms, they can alert the medical team. They should have the ability to observe and report any abnormality. I see this person as a censor who detects problems.* (Obstetrician-Administrator 5, 20 years work experience, hospital 7)

*I want to teach and train the companion. They should learn how to assess the labour pain, where they can check or touch. They will be the one who communicates with the nurses that the contraction is more frequent and intense. They can tell the nurses that the patient wants to push already.* (Labour nurse-midwife 15, 5 years work experience, hospital 6)

If labour companions were trained, for example, during childbirth education classes or antenatal visits, these health workers believed that they could help the woman to manage pain, and communicate to the health workers if the woman needs help or is ready to push.

These critical roles played by labour companions are important facilitators to the *psychological capability* domain of behaviour change, which can influence the relationship between motivation and enacting the behaviour (labour companionship). If labour companions are appropriately equipped with the skills and knowledge to support women during labour, then they in turn have increased motivation, and health workers may feel better able to integrate them into the care team.

### Training the labour companion

Participants expressed that potential labour companions should receive training to understand the process of labour and how to best support the woman. Preparation of the labour companions could be integrated into the existing antenatal classes. Most participants agreed that the training and preparation for the labour companion should start in the third trimester, approximately week 32 of the pregnancy. They should attend the class at least two times, for about 30–60 min. The key content and skills for labour companions to learn during these sessions is how to provide emotional support, pain management techniques and understanding the process of labour. One female participant said that the labour companion should understand the emotions while the woman is going through labour pain so they can support the woman appropriately.

*The labour companion has to learn how to support the patient. We should teach them what labour is and the pain associate to the labour, how much pain, when to report to the medical team. For instance, if the patient's water broke, they have to let us know. If the patient wants to push, they have to report. (Obstetrician 2, 3 years work experience, hospital 7)*

*They have to learn the labour process. It will be somewhat a long process so they can help with the pain while waiting for the delivery. They can be a pushing coach. They have to be perceptive to our moods. (Postpartum woman 24, 21 years, hospital 1)*

The husband of a pregnant woman echoed the desire for learning how to support his wife, and particularly how he could help ease her pain during labour:

*I want to learn what I should do, the process of getting on the labour and delivery wards, what to do when I am on the ward, how I can help my wife with the pain. (Husband 8, 35 years, hospital 1)*

Appropriate training of the labour companion is an important facilitator to the *physical* and *psychological capability* domains of behaviour change, which can increase *motivation*.

### Factors affecting implementation

While all participants noted the many benefits to having a labour companion, some barriers and challenges to implementing companionship were identified. These factors affecting implementation are important barriers and facilitators to *physical* and *social opportunity*, as they relate to creating enabling physical environments and influencing positive sociocultural norms. Many labour and delivery wards in public hospitals are not designed to accommodate labour companions, as the wards are already crowded with women in labour. Consequently, four main barriers were identified by participants: (1) maintaining privacy and confidentiality, (2) increased risk of infection, (3) risk of lawsuits and (4) perceived additional work for health workers to support companions. Maintaining privacy was already a challenge without labour companions, as the labour ward beds are close together, in a narrow and crowded room. In Thai culture, it is improper for women's bodies to be exposed; therefore, if a labour companion is a male, it may be uncomfortable for other women in labour at the same time.

*Our hospital is a public hospital, not a private one. When the patients in labour, waiting to deliver, they are in their bed with a curtain as a divider between beds. There is no privacy. It's difficult for me to work and to protect my patients' privacy. For example, I am trying to do the pelvic exam but the next bed has a husband accompany her. The voices can travel through. It's difficult to work. (Obstetrician 4, 2 years work experience, hospital 4)*

In addition to the challenges of physical privacy, some participants also feared that having more visitors and relatives on the ward will be difficult for the medical team to protect the confidential information of patients.

*I am very afraid of the risk of the confidentiality violation. The companions might talk about other patients to other people. I am very worried about this. (Antenatal nurse-midwife 9, 21 years work experience, hospital 3)*

Participants, particularly healthcare providers, expressed concerns about increased risks of infection, as the ward is usually crowded with women in labour. Adding the labour companion could lead to the increased risk of infection spread (referring to non-COVID-19 infection).

*I think it's kind of risky for the infection. People wear their normal clothing, not sterile. That might increase the infection spread. (Labour nurse-midwife 2, 3 years work experience, hospital 7)*

Healthcare providers expressed concern that the presence of a labour companion may lead to misunderstanding and lawsuits. They worried that while they are providing care, the companions might think that the medical team are disorganised and in chaos, and that people may post these issues on social media. These misunderstandings and miscommunications had the potential to lead to lawsuits.

*When I am on duty, I have to be more careful. My co-workers also warn me about this. For instance, I might be using my smartphone playing on my break but the relatives think I am not helping the patient who are yelling from pain. If they record and pose on social media, people see and misunderstand that I am not doing my job. Having a labour companion is like a two-edged sword. It has good and bad points.* (Labour nurse-midwife 14, 10 years work experience, hospital 6)

Last, many of the study hospitals had high ratios of women to healthcare providers, and healthcare providers feared that introducing companions to the ward may increase their workloads.

*The objective of having a labour companion is to have someone to help us. But I doubt that the person can really help me. I have to explain and communicate more. It will double the communication times because I not only communicate with a patient, I have to communicate with the relatives.* (Labour nurse-midwife 6, 3 years work experience, hospital 2)

For successful implementation of companionship, these barriers would need to be considered and addressed in the implementation strategy. However, despite the barriers, the participants, particularly healthcare providers, believed that the potential benefits of introducing labour companionship would outweigh the risks, suggesting that labour companionship was highly acceptable.

*I think it's possible to implement the labour companion policy because of the substantial benefits. It is easily acceptable. There are many evidence-based research that shows the benefits of having a labour companion can reduce the active and the second phase of the labour, they will change the policy and practice.* (Obstetrician 8, 3 years work experience, hospital 5)

*If there is a policy to include the labour companion, I think it's possible to follow. They have to provide the space. When the direct order comes to the hospital to do it, they will set up more private space. I think it's possible. There shouldn't be any problems.* (Obstetrician-Administrator 1, 35 years work experience, hospital 7)

### Understanding factors affecting implementation using the COM-B model

Figure 1 maps the potential factors affecting implementation from the qualitative interviews and readiness assessment to the COM-B model of behaviour change. The defined behaviour is that all women have the option to have a companion of their choice throughout labour and childbirth. In short, to improve *capability* to have a labour companion, potential labour companions should be well trained and prepared on how to support women throughout labour and birth, and measures may need to be taken to improve privacy. To improve *motivation* to have a labour companion, all stakeholders (women, potential companions and healthcare providers) should be knowledgeable about the benefits of companions and how to

efficiently integrate them into care, and trust-building between healthcare users and healthcare providers may need to take place in contexts with fear of litigation. To improve *opportunity* to have a labour companion, labour wards may need to be physically reorganised to optimise space for a companion and woman to interact, revisions may be needed to allow consistent visitation rights for companions regardless of day or time and facility or public policies may need revision to encourage companionship. To optimise the likelihood for this behaviour to occur in the QUALI-DEC hospitals in Thailand, the implementation strategies should ensure that the key barriers identified are addressed, and that the facilitators are present and encouraged in all sites.

## DISCUSSION

We found that healthcare providers, women and potential companions in eight public hospitals in Thailand had generally positive attitudes towards having labour companions and particularly belief that labour companions would provide beneficial psychological and physical support for the women. However, we identified some opportunities and threats to implementing labour companionship for all women. Training the labour companion, for instance through childbirth education classes or attendance at antenatal visits, was important to ensure that the companion knew how to support the woman and understood what to expect during labour and birth. Limited physical space on the labour wards, overcrowding and multiple beds in the same labour room were major concerns to introducing labour companionship. While policies at the hospital and national level do not currently mention labour companionship, changes are more likely to be made at the hospital-level. For example, current restrictions on the timing of visitations and number of visitors allowed may challenge the implementation of labour companionship and may need to be adjusted prior to implementation to ensure that companions are not subjected to visitor restrictions.

A key facilitator related to the social opportunities is that historically in Thai culture, childbirth occurred at home where the woman was surrounded by her family, and strong values and happiness in welcoming a new family member. Introducing labour companionship for births occurring in health facilities may therefore reflect the values and cultural appropriateness of having a woman's social network supporting her during labour and birth. While there are important barriers to address, namely around policies, training and reorganisation of the physical environment for birth, social opportunities and psychological capabilities that value companionship are critical which appear to be present in Thai culture. These facilitators and barriers are remarkably similar to an implementation study conducted in public hospitals in Egypt, Lebanon and Syria, where women and families highly valued companion support, but health workers identified critical organisational factors such as

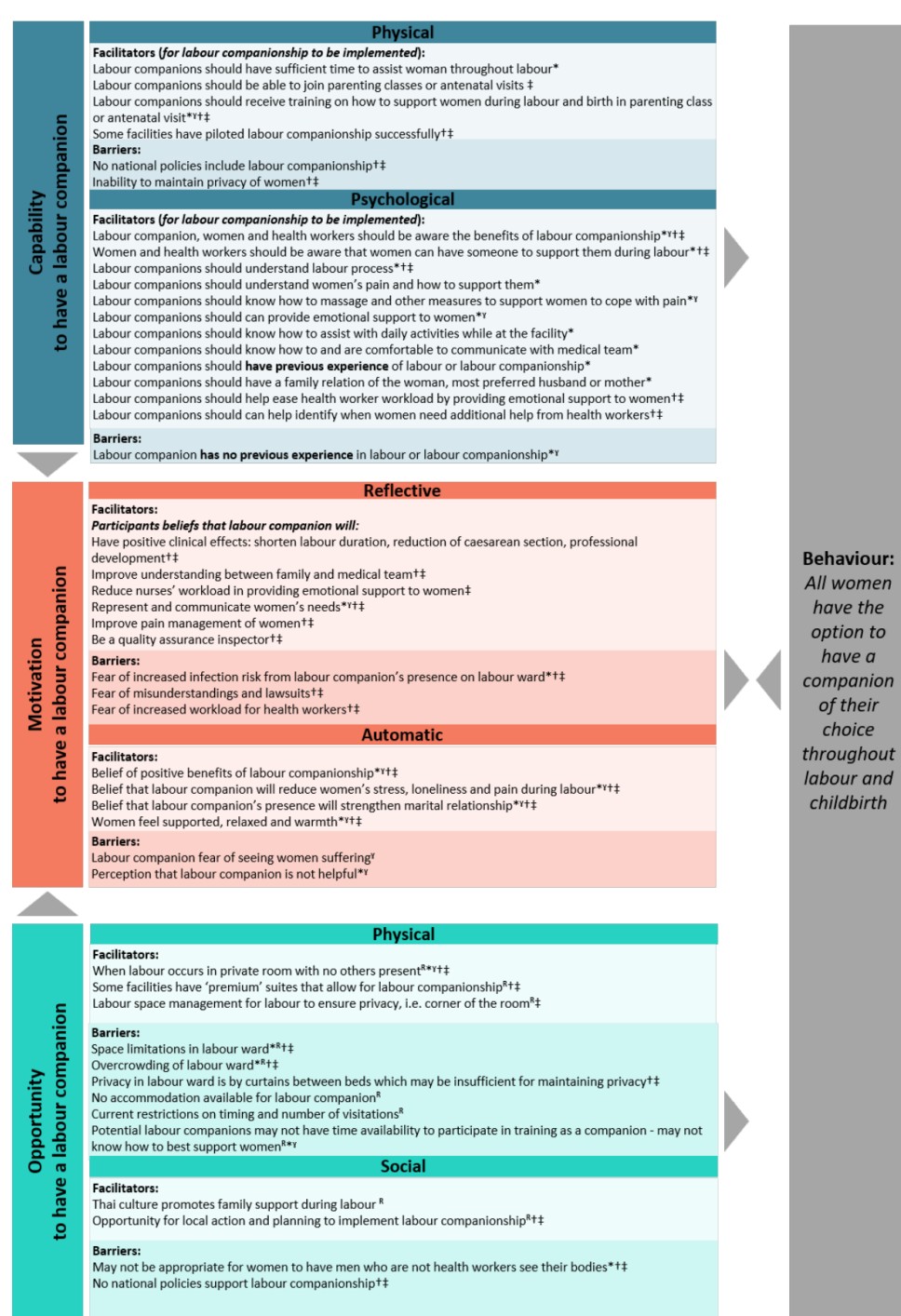

**Figure 1** Mapping the factors affecting implementation of labour companionship in Thailand to the COM-B model of behaviour change. This figure maps the factors affecting labour companionship from the qualitative research findings and readiness assessment to the COM-B model of behaviour change. The COM-B model is a useful way to identify what changes need to occur for an intervention—such as companionship—to be effective. Developing implementation strategies that capitalise on the facilitators and address the barriers to capability, opportunity and motivation is a critical next step for the QUALI-DEC project. Data coming from: *=women, ᵞ=labour companion, †=doctors, ‡=nurse-midwives, R=readiness assessment. COM-B, Capability, Opportunity, and Motivation behaviour change model.

limited physical space, lack of training of companions and limited policy engagement as barriers to successful implementation.[14 15] The implementation study in Egypt, Lebanon and Syria used participatory engagement through engagement with hospital leaders, seminars with healthcare providers, communications materials for companions and changes to the physical space (chairs for companions, curtains around beds, access to hot water and toilets and disposable gowns and nametags for companions) to address these barriers,[14] which may also be a useful approach to inform the QUALI-DEC implementation.

Afulani and colleagues similarly explored women and health workers' perceptions of labour companionship in a public maternity unit in rural Kenya and identified similar facilitators to labour companionship and roles that labour companions could play.[16] In contrast to our findings, the Kenyan study identified additional social barriers, including women's belief that companions cannot help them, embarrassment to have a non-health worker see them during labour and fears that the labour companion would gossip about what they saw during the birth to others or that the labour companion may abuse the woman during labour.[16] While we did not identify these social barriers to implementation, it is possible that particularly the embarrassment and fears of gossip and abuse may be present in more rural areas of Thailand (all QUALI-DEC study hospitals are in urban areas and therefore may not be as influenced by these factors present in smaller communities).

Most women and companions believed a partner or husband to be the optimal companion, believing that witnessing the pain and supporting during the difficult time could strengthen the family bonding including the father and the baby, which was consistent with previous studies.[4 17] Only a few women preferred her mother as a companion. This finding is different from other women in India and Bangladesh, most those women wanted their mothers to be a companion.[18 19] Having a female companion, especially a mother, could yield other benefits, as they can share her own experiences of childbirth, which could serve as encouragement to women. We note that cultural and gender norms may influence the choice of a companion and that ultimately the woman herself should be the person who chooses who will support her.

There are several key implications for research, practice and implementation of the QUALI-DEC study. We plan to use opinion leaders (influential and respected healthcare leaders who are effective communicators and identified by their colleagues or local authorities) at each study hospital to help support implementation.[7] We plan to engage with the opinion leaders during an intensive, 5-day pre-study training workshop, where we will present the results of this formative research and engage to design strategies to optimise implementation.[7] Engaging with the opinion leaders about the benefits of labour companionship and codesigning strategies to address barriers to implementation that are feasible and acceptable in their clinical settings will be critical. For example, we will explore how to assuage healthcare providers' fears that introducing companions will result in higher workloads, potentially through training solutions to help healthcare providers understand benefits of companions and how to integrate them in their care—a similar approach to Kabakian-Khasholian and colleagues.[14] Similarly, we will discuss how to negotiate improving accountability of the health system to women and their families, with the potential risk that instances of poor quality of care are shared on social media by companions.

Moreover, we expect that at a minimum, some reorganisation of the physical space of the labour ward will be needed, for example, introducing chairs, developing plans to mitigate the risk of overcrowding and supplying curtains where necessary to enhance privacy. Likewise, some facility policies may need to be adjusted to change restrictions on visiting hours for the labour ward to ensure that companions are not subjected to visitor restrictions. More work will be needed to explore how to engage with labour companions during the antenatal period, and information, education and communications (IEC) materials are currently being developed to communicate how companions can support women and how health workers can engage them in care. The findings from this study have informed what type of material should be included in IEC materials for women and families, as well as health providers. For example, helping to clarify what to expect from a labour companion, how labour companions can help before, during and after the birth, and practical information to help labour companions support women to the best of their abilities.

Our study had both limitations and strengths. While we aimed to include diverse public hospitals across different regions of Thailand, the findings may not be transferable to all settings in Thailand, including Southern Thailand where we could not include any hospitals. All study hospitals were in urban settings and generally hospitals with relatively high caesarean section rates, so there may be limited transferability to rural settings or settings with lower caesarean section rates. We collected the data during the COVID-19 pandemic, which may have introduced additional barriers to implementation around people's presence on the labour wards (during the data collection period July to October 2020, there were typically less than 10 COVID-19 cases per day in Thailand). We note that WHO COVID-19 clinical management guidance recommends that during the pandemic, all women should have access to woman-centred, respectful care, including a companion of their choice; this includes women with suspected, probably or confirmed COVID-19.[20] Key strengths of our study include triangulation of results from qualitative research and the facility readiness assessment and mapping of key factors affecting implementation to the COM-B model to guide decision-making during QUALI-DEC intervention design and support a systematic, targeted and theory-based development of implementation strategies.

## CONCLUSION

Labour companionship is viewed by women, potential companions and health workers as highly beneficial and acceptable in the Thai context. If labour companions are well-trained on how to best support women, help them to manage pain and engage with healthcare teams, it may be a feasible intervention to implement in the study hospitals. However, key barriers to introducing labour companionship must be addressed to maximise the likelihood of

success. This includes changes to the physical environment in the labour ward to ensure that privacy can be adequately maintained and that there is space for companions to comfortably support women. Facility-level policies may need adjustment, particularly around visitation hours and where companions are not restricted. Context-specific solutions may need to be developed to assuage health worker concerns about potential misunderstandings, lawsuits or reputational risks stemming from the introduction of labour companionship. Health workers will also need training to understand how to engage with labour companionships as part of a woman's care team, to minimise the risk of role encroachment and understand how companionship can be mutually beneficial. These key findings will be considered and deliberated on when developing the QUALI-DEC implementation strategies for introducing labour companionship.

**Author affiliations**
[1]Faculty of Nursing, Khon Kaen University, Khon Kaen, Thailand
[2]Faculty of Nursing, Mahidol University, Bangkok, Thailand
[3]Department of Obstetrics and Gynaecology, Faculty of Medicine, Khon Kaen University, Khon Kaen, Thailand
[4]Gender and Women's Health Unit, Centre of Health Equity, Melbourne School of Population and Global Health, The University of Melbourne, Carlton, Victoria, Australia
[5]Department of Global Public Health, Karolinska Institutet, Stockholm, Sweden
[6]London School of Hygiene and Tropical Medicine, London, UK
[7]Centre Population et Developpement (CEPED), Institute for Research on Sustainable Development, IRD-Université de Paris, ERL INSERM SAGESUD, Paris, France
[8]UNDP/UNFPA/UNICEF/World Bank Special Program of Research, Development and Research Training in Human Reproduction (HRP), Department of Sexual and Reproductive Health and Research, World Health Organization, Geneve, GE, Switzerland

**Acknowledgements** We thank the woman, potential companions and health workers for their valuable time and are grateful to the social scientist team for their hard work in data collection: Nilubon Rujiraprasert, Sasitara Nuampa, Natthapat Buaboon, Waraluk Kittiwatanapaisan and Dasavanh Bounmany. We are indebted to Miss Nampet Jampathong for her professionalism as a project coordinator during the data collection process and her assistance in data transcription.

**Contributors** SR, AR, PL, CH, AD, MdL, APB and MAB designed the study. SR, AR and PL led data collection with support from CH, AD, MdL, APB and MAB. SR led data analysis with support from RIZ and MAB. SR and MAB drafted the manuscript, and all authors reviewed the manuscript. MAB was responsible for the overall content and acted as guarantor.

**Funding** This study is part of the QUALI-DEC-project which is cofunded by the European Union's Horizon 2020 research and innovation programme under grant agreement No 847567 and by UNDP-UNFPA-UNICEF-WHO-World Bank Special Programme of Research, Development and Research Training in Human Reproduction (HRP), a cosponsored programme executed by the World Health Organization (WHO) in the Department of Sexual and Reproductive Health and Research (SRH). MAB is supported by an Australian Research Council Discovery Early Career Researcher Award (DE200100264) and a Dame Kate Campbell Fellowship (University of Melbourne Faculty of Medicine, Dentistry, and Health Sciences).

**Competing interests** None declared.

**Patient and public involvement** Patients and/or the public were not involved in the design, or conduct, or reporting, or dissemination plans of this research.

**Patient consent for publication** Not applicable.

**Ethics approval** This research was approved by the Thai Central Research Ethics Committee (CREC) (COA-CREC020/2020), related university research ethics committees and all hospital research ethics committees. Scientific and technical approval was obtained from the WHO Human Reproduction Programme (HRP) Review Panel on Research Projects (RP2) and ethical approval by the WHO Ethical Review Committee (protocol ID, 004571) and the French Research Institute for Sustainable Development. Participants gave informed consent to participate in the study before taking part.

**Provenance and peer review** Not commissioned; externally peer reviewed.

**Data availability statement** Data are available on reasonable request. Data are available on reasonable request from the corresponding author.

**ORCID iDs**
Pisake Lumbiganon http://orcid.org/0000-0001-9372-0071
Rana Islamiah Zahroh http://orcid.org/0000-0001-7831-2336
Claudia Hanson http://orcid.org/0000-0001-8066-7873
Alexandre Dumont http://orcid.org/0000-0003-3826-0193
Myriam de Loenzien http://orcid.org/0000-0001-7121-0185
Ana Pilar Betrán http://orcid.org/0000-0002-5631-5883
Meghan A. Bohren http://orcid.org/0000-0002-4179-4682

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
