## [Reviewer comments · BMJ Open]

ARTICLE DETAILS

TITLE (PROVISIONAL)	Factors influencing the implementation of labour companionship: formative qualitative research in Thailand
AUTHORS	Rungreangkulkij, Somporn; Ratinthorn, Ameporn; Lumbiganon, Pisake; Zahroh, Rana; Hanson, Claudia; Dumont, Alexandre; de Loenzien, Myriam; Betran, Ana Pilar; Bohren, Meghan

VERSION 1 – REVIEW

REVIEWER	da Matta Machado Fernandes, Luisa Fundação Oswaldo Cruz
REVIEW RETURNED	04-Sep-2021

GENERAL COMMENTS	The study conducted is well designed and has a robust method. Undoubtedly, it will contribute to advance women-centered care during childbirth in Thailand. Without diminishing its merit, I believe it would strengthen the paper if the discussion is further expanded, including papers with a strong qualitative methodology. The paper is unbalanced, with a greater focus on the results and a shorter discussion section. My recommendation is to explore further women's perception of companions and the relevance of a positive childbirth experience. The abstract presented is also unbalanced. The current abstract discussion section should reflect the study discussion, including a comparison with international experience. Currently, the section seems to present the results. However, those can be improved, being faithful to the findings and avoid presetting conditional hypotheses. I believe those adjustments will increase the paper's interest and readability, and the authors will be able to attend to the requests efficiently.
---

REVIEWER	Lobis, Samantha Vital Strategies
REVIEW RETURNED	23-Dec-2021

GENERAL COMMENTS	Overall comments: • I enjoyed reading this manuscript and learning about how the authors are building the foundation for their companionship project in Thailand. The authors' use of both facility assessments and in-depth interviews has given them rich information to draw on when planning their interventions. This manuscript presents a lot of this useful information and I believe with some restructuring and revisions, the findings will become more valuable to implementers in Thailand as well as in other countries. I have provided some line-by-line detailed comments that may be helpful to the authors. I have also made suggestions for the reorganization of some sections. An overall
--

	copyedit would also improve the article.  • Use of 'labour companion': The authors may want to consider specifying which phases of the birth process a labour companion is intended to cover. Is it only for labour or should the companion also be present and provide support at the time of birth and during the postnatal period (pre-discharge)? Introduction:  • Overall: this section should be tightened and made more concise. • L 68-70: Reword so this sentence is clearer • L 78: change 'encouraging her to mobilize' to 'encouraging her to be mobile' • L 108-129: Rewrite this paragraph so that readers better understand the justification of this study (understanding facilitators and barriers to companionship in government hospitals in order to design interventions leading to increased companion use) and how it connects to the larger QUALI-DEC project (appropriate use of caesarean section). Somewhere in the introduction, there needs to be a clearer explanation of the links between continuous support during labour and delivery and how that contributes to the appropriate use of caesarean section (e.g., how experience of care - reduced stress, less disrespect and abuse, more emotional support, greater use of comfort measures, etc. - connects to the more appropriate use of caesarean section). • L 126-129: The objectives of the paper need to be more clearly articulated (e.g., when you say that you are going to 'describe the needs and preferences related to labour companionship' whose needs and preferences? Women's? Health providers? Hospital administrators? Whose behaviors are you looking to change?). Methods:  • Overall:  o It would be useful to more clearly introduce the steps that you took for this paper (e.g., how you conducted IDIs and facility assessments and then synthesized and interpreted the findings from those studies using the COM-B model). This could be done in an opening paragraph to this section. o Consider adding a sub-section called: 'setting' and move the information about the eight hospitals there; consider consolidating / moving any background details about the hospitals from other places to this new sub-section. • L 138: consider clarifying what you mean by 'potential companions (after birth)' – it becomes clearer farther down in the paper but it should be explained here. • L 143-145: combine these sentences to make the point clearer. • L 164-173: Tighten this paragraph; consider dropping the detailed description of all the interview guide topics (because the entire guide is available to readers) and instead focus on the brief description of the labour companionship module. Also, add that this module was used for all types of respondents (pregnant women, postpartum women, companions and health providers). Add what the overall research questions were for this part of the study (in place of adding the specific questions found in the interview guide). • L 174-182: This paragraph should be rewritten to more clearly explain the elements of the readiness survey that relate to companionship (e.g., privacy, crowding and layout). What were the overall research questions for this part of the study? • L 183-193: Who conducted the facility assessment? Was it the same six female nursing professors? • L 195-210: Tighten up this paragraph.
--	---

	 • L 211-224: Explain if you are targeting health providers' behaviors to make them more supportive of companionship and/or if you are trying to change women's behaviors so that they are more likely to bring a companion and/or if you are trying to change potential companion's behaviors so that they are more likely to be companions in subsequent pregnancies. I would imagine that these would need to be analyzed both separately and together. • L 225-237: this should be shortened; some of the information here is repeated elsewhere; some of this information belongs at the end of the article. Results:  • L 239-248: This paragraph can be shortened. For example, L 245: 'the other half were multiparous.' can be deleted • L 254-280 - Contextual insights from readiness assessment: this section can be made more concise by moving the commentary to the discussion section (e.g., L257-258, L264-267, L272-280). Consider adding some findings about staffing – do the hospitals have sufficient staff in the labour wards for all shifts? Also consider adding some findings about whether researchers found any orientation materials, protocols, guidelines, policies etc. at hospitals that would either inhibit women from having companions or facilitate women's ability to have continuous support during childbirth. • Qualitative findings related to labour companionship  o L 286-387 – benefits of labour companions: I think this section has many important points but would benefit from being reorganized. Consider organizing it into the unique benefits perceived by pregnant/postpartum women, unique benefits perceived by potential companions and unique benefits perceived by health providers. You could also have a short paragraph at the start of this sub-section which summarizes the perceived benefits that were universal for all respondent groups. o L 293-296: Consider moving this point to the subsection on the COM-B model o L 400-411: Did any of the respondents express concerns about companions being expected to do too much, taking on too much responsibility for identifying clinical problems or being expected to do things beyond providing continuous emotional/informational/practical support for women, particularly when there are severe staffing shortages? This was an issue that implementers identified and addressed in a birth companionship pilot conducted in Kigoma, Tanzania; implementers defined the roles and limitations of birth companions and conducted routine support visits and implementation research to address any problems that arose (Chaote, P., Mwakatundu, N., Dominico, S. et al. Birth companionship in a government health system: a pilot study in Kigoma, Tanzania. BMC Pregnancy Childbirth 21, 304 (2021). https://doi.org/10.1186/s12884-021-03746-0) o L 415-419: Consider moving this point to the subsection on the COM-B model. o L 424-430: The quote is not needed; the same point is already stated in the preceding text. o L 434-437: The quote is not needed; the same point is already stated in the preceding text. o L 460-465: This text can be shortened. L 461-463: Consider moving this point to the subsection on the COM-B model. o L 470-471: This sentence needs to be reworded to be clearer. o L 490-500: This a finding that should be explored in the discussion section – how to build trust between health providers and women/companions and the importance of strengthening
--	--

	accountability mechanisms.  o L 501-507: This is a finding that should be explored in the discussion section. o L 523-528: This sub-section would be clearer if you broke it down by type of respondent (women, potential companion and health provider) and summarize capabilities, motivations and opportunities for each respondent group. This may help you develop more targeted interventions for health providers vs. women vs. potential companions. Discussion:  • I suggest that you reorganize some of the discussion section around the COM-B model; in this way you could explore priority interventions to address identified barriers and concerns and that take advantage of the many facilitators that were identified in your research (drawing on the literature as you already did). o As indicated above under 'results', I would like to see some discussion on the need for trust and accountability and how that could be addressed. o Based on your findings, it would be useful to add some discussion on the type of preparation that companions will need and how that will be developed. o The need to adapt the physical space to decrease crowding and to increase audio and visual privacy was a major finding from both the facility assessment and IDIs. This should be explored further in the discussion. Conclusion:  • Your conclusions are strong. Once you do some reorganization, the other sections will lead nicely to your conclusions.
--	---

REVIEWER	Singh, Shalini Indian Council of Medical Research, Division of Reproductive Biology and Maternal Health
REVIEW RETURNED	29-Dec-2021

GENERAL COMMENTS	Very important study which reiterates that despite recognizing the benefits of labour companion, the facility birth environment does not support to provide companionship. Similar findings were seen in our study published in Health Policy and Planning, 36, 2021, 1552–1561 titled "Presence of birth companion—a deterrent to disrespectful behaviours towards women during delivery: an exploratory mixed-method study in 18 public hospitals of India"
---

VERSION 1 – AUTHOR RESPONSE

Section	Reviewer comment	Author response
Reviewer 1		
Overall	The study conducted is well designed and has a robust method. Undoubtedly, it will contribute to advance women-centered care during childbirth in Thailand. Without diminishing its merit, I believe it would strengthen the paper if the discussion is further expanded, including papers with a strong qualitative methodology. The paper is unbalanced, with	Thank you for your feedback. Please see revisions throughout the discussion section to address these comments and from the other reviewers.

	a greater focus on the results and a shorter discussion section. My recommendation is to explore further women's perception of companions and the relevance of a positive childbirth experience. The abstract presented is also unbalanced. The current abstract discussion section should reflect the study discussion, including a comparison with international experience. Currently, the section seems to present the results. However, those can be improved, being faithful to the findings and avoid presetting conditional hypotheses. I believe those adjustments will increase the paper's interest and readability, and the authors will be able to attend to the requests efficiently.	
Reviewer 2		
Overall	I enjoyed reading this manuscript and learning about how the authors are building the foundation for their companionship project in Thailand. The authors' use of both facility assessments and in-depth interviews has given them rich information to draw on when planning their interventions. This manuscript presents a lot of this useful information and I believe with some restructuring and revisions, the findings will become more valuable to implementers in Thailand as well as in other countries. I have provided some line-by-line detailed comments that may be helpful to the authors. I have also made suggestions for the reorganization of some sections. An overall copyedit would also improve the article. Use of 'labour companion': The authors may want to consider specifying which phases of the birth process a labour companion is intended to cover. Is it only for labour or should the companion also be present and provide support at the time of birth and during the postnatal period (pre-discharge)?	Thank you for your feedback. Please see revisions throughout to the specific comments.
Introduction	Overall: this section should be tightened and made more concise. L 68-70: Reword so this sentence is clearer L 78: change 'encouraging her to mobilize' to 'encouraging her to be mobile'	Please see revisions throughout the first paragraph to address these comments.
Introduction	L 108-129: Rewrite this paragraph so that readers better understand the justification of this study (understanding facilitators and barriers to companionship in government hospitals in order to design interventions leading to increased companion use) and how it connects to the larger QUALI-DEC project (appropriate use of caesarean section). Somewhere in the introduction, there needs to be a clearer explanation of the links between continuous support	Please see revisions throughout the section "The QUALI-DEC project to address these comments.

	during labour and delivery and how that contributes to the appropriate use of caesarean section (e.g., how experience of care - reduced stress, less disrespect and abuse, more emotional support, greater use of comfort measures, etc. - connects to the more appropriate use of caesarean section).	
Introduction	L 126-129: The objectives of the paper need to be more clearly articulated (e.g., when you say that you are going to 'describe the needs and preferences related to labour companionship' whose needs and preferences? Women's? Health providers? Hospital administrators? Whose behaviors are you looking to change?).	Please see revised text: Based on the formative research conducted among the local stakeholders in Thailand, the aim of this paper is to describe the needs and preferences of women, potential companions, and healthcare providers related to labour companionship, and to map factors that might affect implementation of labour companionship in Thailand, using a behaviour change model.
Methods	It would be useful to more clearly introduce the steps that you took for this paper (e.g., how you conducted IDIs and facility assessments and then synthesized and interpreted the findings from those studies using the COM-B model). This could be done in an opening paragraph to this section. o Consider adding a sub-section called: 'setting' and move the information about the eight hospitals there; consider consolidating / moving any background details about the hospitals from other places to this new sub-section. •	Please see the revised text: This is a formative qualitative study using a health facility readiness assessment and in-depth interviews (IDIs) with women, potential companions, and healthcare providers, described in detail in the study protocol (9) and below. In short, the readiness assessment and IDIs explored the needs and preferences of these key stakeholders to introduce labour companionship in each setting. During the analysis, we conceptualised findings from the readiness assessment and IDIs as 'factors potentially affecting implementation of labour companionship', and used behaviour change frameworks to map the findings in order to better understand what is needed to develop effective intervention implementation strategies. We note that we have left the details about the setting in the general methods section as it is a short paragraph (2 sentences) that does not warrant a separate section.
Methods	L 138: consider clarifying what you mean by 'potential companions (after birth)' – it becomes clearer farther down in the paper but it should be explained here. L 143-145: combine these sentences to make the point clearer.	Please see clarification in the 'Participants and recruitment' : Five groups of participants were identified for this study: 1) pregnant women, 2) postpartum women, 3) a person identified by the woman as someone she would have liked as a companion (potential companions; before birth), 4) potential companions (after birth), and 5) healthcare providers (doctors, nurse-midwives)

		and administrators or managers. Please see combined sentence (line 160 revised version)
Methods	L 164-173: Tighten this paragraph; consider dropping the detailed description of all the interview guide topics (because the entire guide is available to readers) and instead focus on the brief description of the labour companionship module. Also, add that this module was used for all types of respondents (pregnant women, postpartum women, companions and health providers). Add what the overall research questions were for this part of the study (in place of adding the specific questions found in the interview guide).	We have dropped the detailed description of the interview guide.
Methods	L 174-182: This paragraph should be rewritten to more clearly explain the elements of the readiness survey that relate to companionship (e.g., privacy, crowding and layout). What were the overall research questions for this part of the study?	Thank you for the feedback. We have not revised this paragraph, as the aim of the paper (including both IDI and readiness assessment analysis) is stated in the last sentence of the introduction, and the elements of the readiness assessment related to companionship are already clearly listed at the end of the paragraph.
Methods	L 183-193: Who conducted the facility assessment? Was it the same six female nursing professors?	We have clarified this: Readiness assessments were conducted by members of the QUALI-DEC research team who were professors of nursing, but not employed by the study hospitals.
Methods	L 195-210: Tighten up this paragraph. L 211-224: Explain if you are targeting health providers' behaviors to make them more supportive of companionship and/or if you are trying to change women's behaviors so that they are more likely to bring a companion and/or if you are trying to change potential companion's behaviors so that they are more likely to be companions in subsequent pregnancies. I would imagine that these would need to be analyzed both separately and together.	Please see revisions throughout the 'Data analysis section' to address these comments.
Methods	L 225-237: this should be shortened; some of the information here is repeated elsewhere; some of this information belongs at the end of the article.	We have moved this section ('Ethical considerations') to the end matter of the manuscript. We defer to the editorial team to assess the appropriateness of this move.
Results	L 239-248: This paragraph can be shortened. For example, L 245: 'the other half were multiparous.' can be deleted	Please see revisions to this paragraph (first paragraph of the results) to streamline.
Results	L 254-280 - Contextual insights from readiness assessment: this section can be made more concise by moving the commentary to the discussion section (e.g., L257-258, L264-267, L272-280). Consider adding some findings about staffing – do the hospitals have sufficient staff in the labour wards for all shifts? Also consider	Please see revisions throughout this section ('Contextual insights from the readiness assessment'). We have moved the interpretation of the readiness assessment findings to the discussion section. We were not able to assess from the

	adding some findings about whether researchers found any orientation materials, protocols, guidelines, policies etc. at hospitals that would either inhibit women from having companions or facilitate women's ability to have continuous support during childbirth.	readiness assessments whether there were sufficient staff on each shift, as this can vary by day, and by time of day, as would be the case in all hospitals. The materials that currently exist and may inhibit women from having companions are discussed – this relates to the visiting hours allowable for women in labor, and may be a barrier given that they are not 24/7. This is already presented in this section and discussed in further detail in the discussion section.
	Qualitative findings related to labour companionship:  1. L 286-387 – benefits of labour companions: I think this section has many important points but would benefit from being reorganized. Consider organizing it into the unique benefits perceived by pregnant/postpartum women, unique benefits perceived by potential companions and unique benefits perceived by health providers. You could also have a short paragraph at the start of this sub-section which summarizes the perceived benefits that were universal for all respondent groups. 1. L 293-296: Consider moving this point to the subsection on the COM-B model 1. L 400-411: Did any of the respondents express concerns about companions being expected to do too much, taking on too much responsibility for identifying clinical problems or being expected to do things beyond providing continuous emotional/informational/practical support for women, particularly when there are severe staffing shortages? This was an issue that implementers identified and 	Thank you for your feedback on the section "Qualitative findings related to labour companionship". Please find responses to each comment below:  1. Where the perspectives of the different types of participants (women, companions, providers) were similar, we have merged these sections together, and where they are different, we highlight the differences. We have chosen this method of organizing the results to reduce duplication. For instance, in the 'Benefits of labour companions' section, the benefits identified by participants were similar across participant groups, so we report them together; we have added this to the first sentence of this section. 1. We have intentionally linked the more descriptive qualitative findings (e.g. preferences and beliefs) to the more analytic findings (mapped to COM-B framework) to a) reduce duplication across the results section, and b) help a reader to understand how these qualitative findings can be considered as 'factors affecting implementation'. 1. Thank you for this comment, we acknowledge that this is a well documented issue in

	addressed in a birth companionship pilot conducted in Kigoma, Tanzania; implementers defined the roles and limitations of birth companions and conducted routine support visits and implementation research to address any problems that arose (Chaote, P., Mwakatundu, N., Dominico, S. et al. Birth companionship in a government health system: a pilot study in Kigoma, Tanzania. BMC Pregnancy Childbirth 21, 304 (2021). https://doi.org/10.1186/s12884-021-03746-0) 1. L 415-419: Consider moving this point to the subsection on the COM-B model. 1. L 424-430: The quote is not needed; the same point is already stated in the preceding text. 1. L 434-437: The quote is not needed; the same point is already stated in the preceding text. 1. L 460-465: This text can be shortened. L 461-463: Consider moving this point to the subsection on the COM-B model. 1. L 470-471: This sentence needs to be reworded to be clearer. 1. L 490-500: This a finding that should be explored in the discussion section – how to build trust between health providers	other settings, in terms of a barrier to introducing companionship as a fear that companions will be asked to do too much. In our study, the research participants did not document this as a potential barrier, and did not mention any fears that a labor companion may be expected to do things beyond providing emotional and practical support for their woman – there were no suggestions that their roles could be substituted for staffing/health workforce shortages. 1. Please see point 2 above. 1. We have removed this quote (section: Training the labour companion). 1. We have removed this quote (section: Training the labour companion). 1. Please see point 2 above. 1. Please see revised sentence “In Thai culture, it is improper for women’s bodies to be exposed; therefore, if a labour companion is a male, it may be uncomfortable for other women in labour at the same time.”
--	--	--

	and women/companions and the importance of strengthening accountability mechanisms. L 501-507: This is a finding that should be explored in the discussion section. 1. L 523-528: This subsection would be clearer if you broke it down by type of respondent (women, potential companion and health provider) and summarize capabilities, motivations and opportunities for each respondent group. This may help you develop more targeted interventions for health providers vs. women vs. potential companions.	1. Please see revisions throughout the Discussion, particularly in paragraphs 5 and 6, to address these comments. 1. Please see added text to summarize figure 1: In short, to improve capability to have a labour companion, potential labour companions should be well trained and prepared on how to support women throughout labour and birth, and measures may need to be taken to improve privacy. To improve motivation to have a labour companion, all stakeholders (women, potential companions, and healthcare providers) should be knowledgeable about the benefits of companions and how to efficiently integrate them into care, and trust-building between healthcare users and healthcare providers may need to take place in contexts with fear of litigation. To improve opportunity to have a labour companion, labour wards may need to be physically reorganised to optimisespa n style="font-family:Calibri; font-style:italic"> space for a companion and woman to interact, revisions may be needed to allow consistent visitation rights for companions regardless of day or time, and facility or public policies may need revision to encourage companionship.
Discussion	 • I suggest that you reorganize some of the discussion section around the COM-B model; in this way you could explore priority interventions to address identified barriers and concerns and that take advantage of the many facilitators that were identified in your research (drawing on the literature as you already did).  o As indicated above under ‘results’, I would like to see some discussion on the need for trust and accountability and how that could be addressed. o Based on your findings, it would be useful to add some discussion on the type of preparation that companions will need and how that will be developed. 	Please see revisions throughout the Discussion, particularly in paragraphs 5 and 6, to address these comments.

	o The need to adapt the physical space to decrease crowding and to increase audio and visual privacy was a major finding from both the facility assessment and IDIs. This should be explored further in the discussion.	
Discussion	Conclusion: Your conclusions are strong. Once you do some reorganization, the other sections will lead nicely to your conclusions.	Thank you.
Reviewer 3		
Overall	Very important study which reiterates that despite recognizing the benefits of labour companion, the facility birth environment does not support to provide companionship. Similar findings were seen in our study published in Health Policy and Planning, 36, 2021, 1552–1561 titled "Presence of birth companion—a deterrent to disrespectful behaviours towards women during delivery: an exploratory mixed-method study in 18 public hospitals of India"	Thank you so much for sharing this very timely new paper! We have read with interest and integrated into our manuscript.